Mesquite bugs, other insects, and a bat in the diet of pallid bats in southeastern Arizona

Czaplewski Nicholas J. nczaplewski@ou.edu 1
Menard Katrina L. 2
Peachey William D. 3
1 Section of Vertebrate Paleontology, Oklahoma Museum of Natural History , Norman , OK , United States of America
2 Section of Recent Invertebrates, Oklahoma Museum of Natural History , Norman , OK , United States of America
3 Sonoran Science Solutions , Tucson , AZ , United States of America
Vonk Jennifer
Electronic publication date: 2018 Dec 4
Publication date: 2018
Volume: 6
Electronic Location ID: e6065
Received 2018 May 17; Accepted 2018 Nov 5
Copyright: ©2018 Czaplewski et al.
Copyright year: 2018
Copyright holder: Czaplewski et al.
License: This is an open access article distributed under the terms of the Creative Commons Attribution License, which permits unrestricted use, distribution, reproduction and adaptation in any medium and for any purpose provided that it is properly attributed. For attribution, the original author(s), title, publication source (PeerJ) and either DOI or URL of the article must be cited.
License URL: https://creativecommons.org/licenses/by/4.0/

Keywords: Biodiversity, Trophic relationships, Diet, Foraging, Antrozous pallidus, Thasus neocalifornicus, Mesquite bug, Pseudokarst, Night roost, Choeronycteris mexicana

Funding: The OMNH Director Michael Mares Departments of Vertebrate Paleontology Recent Invertebrates The Arizona Game and Fish Commission I-96-024 The OMNH Director Michael Mares and the departments of Vertebrate Paleontology and Recent Invertebrates provided financial support for our research. The Arizona Game and Fish Commission provided a grant (I-96-024) that lead to the initial discovery of the roost with culled insect remains. There was no additional external funding received for this study. The funders had no role in study design, data collection and analysis, decision to publish, or preparation of the manuscript.

==============================
The pallid bat (Antrozous pallidus) is a species of western North America, inhabiting ecoregions ranging from desert to oak and pine forest. They are primarily insectivorous predators on large arthropods that occasionally take small vertebrate prey, and are at least seasonally omnivorous in certain parts of their geographic range where they take nectar from cactus flowers and eat cactus fruit pulp and seeds. Until recently, mesquite bugs were primarily tropical-subtropical inhabitants of Mexico and Central America but have since occupied the southwestern United States where mesquite trees occur. Using a noninvasive method, we investigated the bats’ diet at the Cienega Creek Natural Preserve, Arizona, by collecting food parts discarded beneath three night roosts in soil-piping cavities in a mesquite bosque. We also made phenological and behavioral observations of mesquite bugs, Thasus neocalifornicus, and their interactions with the mesquite trees. We determined that the bats discarded inedible parts of 36 species in 8 orders of mainly large-bodied and nocturnal insects below the night-roosts. In addition, one partial bat wing represents probable predation upon a phyllostomid bat, Choeronycteris mexicana. About 17 of the insect taxa are newly reported as prey for pallid bats, as is the bat C. mexicana. The majority of culled insect parts (88%) were from adult mesquite bugs. Mesquite bug nymphs did not appear in the culled insect parts. After breeding in late summer, when nighttime low temperatures dropped below 21 °C, the adult bugs became immobile on the periphery of trees where they probably make easy prey for opportunistic foliage-gleaning pallid bats. Proximity of night-roosts to mesquite bug habitat probably also enhances the bats’ exploitation of these insects in this location.

Introduction

The pallid bat, Antrozous pallidus, is widespread in western North America in arid to semiarid, rocky habitats, open deserts, and oak and pine forest. Previous studies have shown that this species is generally a predator on arthropods, especially insects. However, pallid bats are opportunistic and flexible, occasionally taking fruit from organ pipe cactus (Stenocereus thurberi) and cardón (Pachycereus pringlei) in the Southwest, and at least incidentally, they also take pollen and nectar from flowering columnar cacti (Howell, 1980; Herrera, Fleming & Findley, 1993; Simmons & Wetterer, 2002; Frick, Heady & Hayes, 2009; Frick et al., 2013; Frick et al., 2014; Aliperti et al., 2017) and probably agaves (Ammerman et al., 2012). Most often, pallid bats prey upon relatively large, flightless arthropods; occasionally they also eat small vertebrates (Engler, 1943; Orr, 1954; O’Shea & Vaughan, 1977; Bell, 1982; Lenhart, Mata-Silva & Johnson, 2010; Rambaldini & Brigham, 2011). Some of these prey items are taken during a brief touchdown or are gleaned from foliage during flight. Pallid bats use a characteristic searching flight that usually involves relatively slow and maneuverable flying about 0.5–2.5 m above the ground while making rhythmic rises and dips interspersed with swoops and glides when the bat detects prey (O’Shea & Vaughan, 1977). Occasionally the bats hover near low or thorny vegetation, or land on the ground where they are quite agile at using a variety of gaits and strides to pursue prey. This foraging style carries high risks of injury and predation for the bats, whose wing membranes and bones heal but show scars and deformities (Davis, 1968). While preying upon scorpions and centipedes they also endure venomous stings to the face and other body parts (Hopp et al., 2017). Previous authors (O’Shea & Vaughan, 1977) showed that pallid bats eat arthropods that share at least two of four characteristics: (1) large size; (2) either obligatorily or primarily active on the ground surface; (3) fly weakly at low heights; or (4) fly strongly but often land on vegetation. The bats frequently retreat to a night roost to rest or manipulate and eat the prey they have caught. They alight on the ceilings of rock shelters, overhangs, or small grottos temporarily to process their prey. The bats drop undesired parts of the arthropods such as wings, elytra, and legs. The discarded items provide qualitative data on pallid bat diets (e.g.,  Orr, 1954; Ross, 1961; Ross, 1967; O’Shea & Vaughan, 1977; Bell, 1982; Lenhart, Mata-Silva & Johnson, 2010), although Johnston & Fenton (2001) found that the insects represented in the culled parts were biased toward the hardest and largest prey species eaten relative to species represented in fecal pellets. In this study, we investigated food habits of pallid bats in southeastern Arizona and made observations on their interactions with the predominant insect in their local diet, the mesquite bug Thasus neocalifornicus. We add to the list of arthropod and vertebrate prey known to be taken by pallid bats. Importantly, we document observations relevant to an example of these bats feeding opportunistically on large numbers of mesquite bugs, as well as the first record in the wild of their feeding on another species of bat in southeastern Arizona. We also provide observations of certain mesquite bug behaviors that may increase their susceptibility to the foraging style of pallid bats.

As gleaning bats that hunt arthropods moving or resting on plant or ground surfaces, pallid bats have well-developed acoustical, olfactory, and visual senses. They mainly locate their prey by sound, either through active echolocation or passive detection of the faint sounds made by moving prey (O’Shea & Vaughan, 1977). In experiments with insects and their pheromones other than those of mesquite bugs (Thasus spp.), pallid bats are also sensitive to olfactory cues and are able to distinguish prey odors from controls and from non-prey species (Johnston, 2002). The bats show selectivity upon closely approaching certain prey insects versus a non-prey noxious insect (the Pinacate beetle Eleodes: Tenebrionidae) or paper balls impregnated with their odors (Johnston, 2002). The bats also have relatively large eyes and high visual acuity at low light levels (Bell & Fenton, 1986).

Pallid bats previously were reported to take certain leaf-footed bugs (Heteroptera: Coreidae) only occasionally. Coreids are globally distributed but mostly subtropical and tropical insects, with about 80 species in the continental United States and Canada (Froeschner, 1988). One member genus, Thasus (Coreinae: Nematopini), has eight species primarily distributed in the Neotropics (Forbes & Schaefer, 2003). Like many other Hemiptera, Coreidae are herbivores that suck the contents of plant tissues (Froeschner, 1988). Many coreids are also host specific, feeding on one or two families of plants (Froeschner, 1988). The only species of Thasus in the United States is Thasus neocalifornicus (giant mesquite bug); the species also occurs in Baja California and Sonora and Chihuahua, Mexico (Forbes & Schaefer, 2003). The closely related species Thasus gigas and Thasus acutangulus occur further south in Mexico and Central America; these two species were once considered synonymous with T. neocalifornicus but have since been shown to be distinct (Brailovsky & Barrera in Brailovsky et al., 1995; Forbes & Schaefer, 2003). Thasus neocalifornicus is ecologically tied to mesquites (Prosopis, Fabaceae; De La Torre-Bueno, 1945; Ward et al., 1977; Schuh & Slater, 1995; Brummermann, 2010). Mesquite trees have spread widely in the United States during the last two centuries along with cattle (Turner et al., 2003), providing the potential for the bugs to expand their geographic range. Schaefer & Packauskas (1997) speculated that the United States populations of T. neocalifornicus in Arizona might have been an accidental introduction by humans. The species has been recorded in Arizona since at least 1876 (Forbes & Schaefer, 2003) and is now also known north of Mexico from California to Texas (https://bugguide.net/node/view/20163).

As a brief synopsis of the known ecology and annual phenological cycle of T. neocalifornicus relative to their host plant, mesquite bugs are univoltine (having one generation per year and overwintering as eggs; (Jones, 1993). Eggs eclose in February, and nymphs aggregate around the eggs to use up the rest of the egg reserves and feed. Nymphs aggregate using specialized pheromones (adults do not respond to the pheromones in tests), and secrete malodorous, toxic compounds in self-defense and possibly as alarm chemicals to alert conspecifics against predator attacks. The nymphal toxins are effective on insect predators in tests; tests do not seem to have been made on vertebrate predators (Prudic, Noge & Becerra, 2008). Nymphs feed on mesquite leaves and pods (once available), and molt through their first-fifth instars from January–July; they often migrate to the base of their host trees in summer, probably in response to high afternoon temperatures (Jones, 1993). Mesquites flower from February-March, and bear fruits (bean pods) from July–October. As nymphs, mesquite bugs are unable to fly and are aposematically colored red, white, and brownish or blackish to advertise their noxious secretions. The coloration probably deters visually oriented diurnal predators such as birds and larger invertebrate predators. Although the warning coloration might be visible to bats during twilight hours, night-active bats might be repelled by the nymphs’ noxious secretions. The nymphs often stay under the foliage on the spiny branches of the mesquite canopy; they also form defensive aggregations that secrete noxious fluid from their abdomens, similar to a related species, T. acutangulus in Central America (Aldrich & Blum, 1978). The adults first start to appear from the fifth nymphal instars around July–August, with the highest proportion of adults between May–September. By contrast with the nymphs, the adult mesquite bugs are large and dark colored (blackish brown and dark reddish) and no longer secrete the chemicals that are toxic to small insect predators. Instead, the adults secrete a different set of noxious chemicals from those of the nymphs to trigger aggregations and in response to a simulated predator disturbance. These chemicals include hexyl acetate, hexenal, and hexanol (Prudic, Noge & Becerra, 2008; Noge, 2015). Adult pheromones are not toxic or deterrent to insect predators but might deter vertebrates such as birds, a major group of predators on adult heteropteran insects (Prudic, Noge & Becerra, 2008). Adult mesquite bugs feed in the mesquite trees until late summer, then breed and lay eggs on mesquite stems and under bark in late summer-early fall. Females start ovipositing around August and continue through October, when the mesquite trees start dropping their leaves (October–January). They overwinter only in the egg stage (Jones, 1993).

During 1994–1996, in the process of studying bats roosting in soil-piping cavities in southern Arizona (Van de Water & Peachey, 1997), we observed reddish guano and culled wings of mesquite bugs beneath a pallid bat night roost. Collecting these discarded fragments eventually grew into the present contribution to knowledge of the behavior of mesquite bugs and the diet of pallid bats.

Materials and Methods

In the process of observing and studying Mexican long-nosed bats at the Cienega Creek Natural Preserve under Arizona Game and Fish Commission permit I-96-024, one of us (WDP) discovered guano and insect parts beneath a night roost of pallid bats in one of several soil-piping cavities. Realizing the potential of these discarded remains to bolster knowledge of the diet of pallid bats in this area, we searched for other such night feeding roosts in the local area and opportunistically revisited them to collect the prey remains while the bats were absent.

Study area

A small remnant mesquite bosque (bottomland forest) occurs at 1,030–1,060 m elevation in an abandoned meander of Cienega Creek, in the Cienega Creek Natural Preserve (CCNP), southeast of Tucson in Pima County, Arizona. Dominant plants in the bosque are velvet mesquite (Prosopis velutina) and graythorn (Zizyphus). Adjacent to the bosque, Cienega Creek flows above ground for parts of its reach where there are surface outcroppings of porphyritic andesite at the upstream and downstream limits of the reach. There the creek forms a riverine marsh or ciénaga, one of few remaining perennial reaches of the stream, and a disappearing habitat feature in the desert southwest (Turner, 1974; Hendrickson & Minckley, 1984). The riparian area is dominated by tree species such as cottonwood (Populus), willow (Salix), ash (Fraxinus), mesquite, and the shrubs seepwillow (Baccharis) and sumac (Rhus). Cattails (Typha) grew in the water of the ciénaga. The bosque grows on a low Quaternary terrace 2–10 m above the stream channel and abruptly separated from it by vertical banks. On adjacent gravelly hills is semi-desert grassland and desert scrub with palo verde (Cercidium) and saguaro (Carnegiea), accented by species of yucca (Yucca), agave (Agave), acacia (Acacia), and ocotillo (Fouquieria), with occasional juniper (Juniperus). Foothills of the Rincon Mountains occur to the north and bear junipers and oak woodland at higher elevations.

The relative representation of trees and many other plants in this area was strongly changed in historic times after colonization; the extensive removal in the 1800s of oaks and junipers for railroads and livestock overgrazing resulted in an increase in the density of mesquites (Bahre & Hutchinson, 1985; Turner et al., 2003). As a result, Cienega Creek became entrenched and presently flows at a lower level than it did during and prior to the 1800s. The lowering of the water table, headward erosion, and subsurface withdrawal are removing the soil beneath the mesquite trees, exposing their roots, gullying the terraces and bosque, and forming a pseudokarst terrain with natural bridges, blind and interrupted reaches, sinkholes, and underground cavities through soil-piping action (Fig. 1). In CCNP, the cavities provided roosting sites for night-roosting pallid bats in the summer time, as well as refuges or nesting areas at various times of year for other species including other bats (Choeronycteris mexicana, Corynorhinus townsendii, and Myotis velifer), woodrats (Neotoma albigula), javelinas (Pecari tajacu), skunks (Conepatus leuconotus and Mephitis macroura), and a small unidentified bird (personal observations). In the immediate vicinity of the soil-piping cavities, plants included mesquite, graythorn, desert broom (Baccharis), cholla cactus (Cylindropuntia), grasses, and small herbaceous plants. The vegetation is essentially the same on top of the flat terrace as in the bottoms of the eroding gullies, except that mesquites are absent in the gully systems. As the soil continues to erode, the soil-piping cavities seem to be ephemeral and might eventually disappear as roosting areas for bats. Episodic roof collapse from the ceilings of the soil-piping cavities occasionally covered the insect parts dropped by the bats.

Figure 1 Plate of four photos (A–D), showing soil-piping cavities, culled insect parts, and bat guano.

(A and B) Two soil-piping cavities developed in the terrace supporting a mesquite bosque on top with mesquite roots being exposed, and grasses in the bottoms, at Cienega Creek Natural Preserve, Arizona. The cavities serve as shelters for a variety of mammals including several species of bats. Cavity in a is dark spot in center of image; cavity in B formed a temporary natural arch. (C) Interior of one of the soil-piping cavities showing a scattering of culled insect parts dropped beneath a night roost of Antrozous pallidus. (D) Close-up view of the scattering; note large numbers of reddish guano pellets (especially within the spotlight from photographer’s headlight at lower left), colored by the contents of mesquite bugs, numerous mesquite bug exoskeletal parts, moth wings, and beetle elytra. Photos A and B by WD Peachey; photos C–E by NJ Czaplewski.

In September 2002, we observed mesquite bugs on and under mesquite trees along a normally dry tributary of Cienega Creek that had flooded the previous night during a rainstorm. We also made casual observations of pallid bats in 2001 and 2002 at a day roost that was discovered in the porch of a caretaker’s residence at the nearby Colossal Cave Mountain Park. This building roost was about 5 km distant from the soil-piping cavities at CCNP and at an elevation of 1,095 m, about 60 m higher than the soil-piping cavities. Although this porch served mainly as a day roost, it was also sometimes used as a night roost by pallid bats.

Collecting methods

Insect parts were first noted by one of us (WDP) in 1994 in one of the soil-piping cavities at Cienega Creek. In 1996, WDP discovered two additional cavities with accumulations of insect fragments and made sightings of A. pallidus. We chose to study the culled insect parts discarded by the pallid bats as a non-intrusive method of determining the macro-arthropodophagous diet in this bat population. Pallid bats are sensitive to disturbance at their roosts (Arroyo-Cabrales & De Grammont, 2017; O’Shea & Vaughan, 1977), thus we collected insect remains at a night roost not used by the bats during the daytime as a way to avoid interference in their activity. The pallid bats were usually absent from the soil-piping cavities when we collected samples in the daytime except on one occasion in September 2002 when we observed two individuals. We visited the cavities and collected insect parts once in November 1996, once in January 2001, twice, in February and September 2002, and once in September 2004. These did not represent seasonal samples but were merely times at which we were able to visit the soil-piping cavities and collect the remains that had accumulated since our previous visit. We collected all pieces from the larger concentrations of pallid bat prey that could easily be picked up by hand for later identification and to sample the overall diversity of species eaten, but also to estimate the relative abundance in the diet of the different insect species. Although this method possibly misses some smaller insects taken in aerial hawking flight (not a preferred mode of foraging for pallid bats; O’Shea & Vaughan, 1977; Johnston & Fenton, 2001), our study reinforces previously published data about the contribution of prey brought into night roosts to the total diet of pallid bats. We identified insect parts by comparison with intact museum specimens in the Oklahoma Museum of Natural History, Section of Recent Invertebrates, with descriptions in the literature, and with digital images and relevant data archived online (e.g., http://www.Bugguide.net). Individual prey parts and specimens collected in this study will be accessioned into the Department of Recent Invertebrates at the Sam Noble Oklahoma Museum of Natural History, where the data will be cataloged and made freely available to the public through GBIF and iDigBio online portals.

On 11 September 2002 we made observations and photographed mesquite bug behavior in the mesquite bosque in late afternoon and early evening. We observed apparent end-of-season mating and mortality of adult insects. We also recorded air temperatures and relative humidity with a handheld electronic sensor outside one of the pallid bat night-roosting cavities during the sundown-to-dark transition period to investigate the relationship of temperature and humidity on adult activities late in the season. On the same date, we collected several of the dead and dying mesquite bugs as voucher specimens. We inferred the sex of the mesquite bugs eaten using sexually dimorphic hind leg parts (Schuh & Slater, 1995).

Results

Among the insects observed at CCNP, mesquite bugs were common in the bosque, active and feeding on mesquites. They followed the typical phenological cycle for tropical areas described in the Introduction. We observed mesquite bugs as nymphs only early in the warm season (Fig. 2). By late summer and early fall, all Thasus observed in the ciénaga area were adults. In late summer the mesquite bugs could be seen flying all over the bosque, alighting on the trees, and mating. When summer heat slowed, evapotranspiration was high, and there was a marked diurnal-nocturnal temperature shift. Cooler air drains from the nearby mountains and a strong down-canyon breeze flows into the bosque and ciénaga. By the end of September and early October, the bugs continued mating but appeared to be succumbing to end of season mortality, possibly due to intolerance of the decreasing nighttime temperatures. At this time of year they became inactive at night and remained exposed on the periphery of the canopy. On 11 September 2002, after the first few nights during which the temperatures started to drop below about 21 °C and the bugs were clustered out on the edges of the branches, we observed individuals become immobile while mating, laying eggs, dying, and falling to the ground (Fig. 2). During the sundown-to-dark transition period on this same date, air temperature decreased by 4.2 °C, from 25.2° to 21.0°, while relative humidity increased by 16%, from 69% to 85%. Upon examination, the fallen bugs on the ground beneath mesquites occurred singly or sometimes in mated pairs (one male and one female in each case). We collected three pairs of the dead ones off the ground as voucher specimens.

Figure 2 Plate of several photos (A–E), showing mesquite bug nymph, adult, mating adults, clustered adults on mesquite, dead adults on the ground.

Mesquite bugs, Thasus neocalifornicus, at the Cienega Creek Natural Preserve, Arizona. (A) T. neocalifornicus nymph (5th instar), with aposematic coloration indicating its noxious nature. (B) Adult, not to same scale as nymph. (C) Adults mating on a mesquite branch at dusk (with flash). (D) Breeding adults clustered on the peripheral foliage of mesquite at dusk in September 2002 (with flash). (E) Scattered dead adults on the ground representing a <24-hour accumulation after a rainstorm had swept away other debris. Photos by NJ Czaplewski.

By 29 September 2002, no live adult mesquite bugs were present in the vicinity of the roosting cavities of CCNP. This was probably due to cold air drainage through the bottomland, because live adults were active in nearby upland areas on the same date. On this date, pallid bats also night-roosted on the caretaker’s building porch in the upland, and many moth wings but no mesquite bug parts were observed beneath the bats. On the night of 1 October 2002, the bats were again present in the porch roost but no new culled insect parts appeared, and the number of pallid bats dwindled until 7 October when only 1 or 2 were present, and no guano was present.

Pallid bats used only three of six available soil-piping cavities in the CCNP mesquite bosque as night roosts during our study, although the other three cavities were sometimes utilized by other species of bats, especially Choeronycteris mexicana (Mexican long-tongued bat) in the summer. The soil-piping cavities (Fig. 1) offered several characteristics that make them suitable as night roosts for pallid bats: (1) enclosed space providing protection from the weather and nocturnal flying predators; (2) easy access with from one to three entrances of relatively large dimensions; (3) relatively spacious interior (in this aspect the cavities were somewhat like the daytime roosts described by Vaughan & O’Shea, 1976) mostly unobstructed except for occasional exposed mesquite roots; (4) high ceilings and steep walls, providing safety from ground and climbing predators, respectively; (5) rough ceiling surface texture providing secure grip for the bats’ claws while processing prey; (6) proximity to the bats’ foraging area, and to at least one observed day roost.

Within the soil-piping cavities, the insect pieces dropped by the bats were concentrated across a small area of the floor on clods of collapsed soil that had fallen from the ceiling (Figs. 1C and 1D). Large guano pellets, often stained red from the mesquite bugs, attributable to the pallid bats occurred within the concentrations of culled insect parts on the floor of the cavities. Uncommon and scattered insect parts were occasionally found distant from these concentrations in the same soil-piping cavities, and probably represented prey remains culled by other species of bats. Three other species of bats were observed using the cavities rarely. Two of these were smaller species than Antrozous pallidus (which has a body weight of 20–35 g; Harvey, Altenbach & Best, 2011). On one occasion, we observed four cave myotis, Myotis velifer (12–15 g) clustered in a small soil pipe in the ceiling not far from one of the cavities used by A. pallidus. On two consecutive days in January 2001 in a different area we observed an individual of Townsend’s big-eared bat, Corynorhinus townsendii (8–14 g), in hibernation. Because the isolated insect fragments could have represented feeding by these other species, they were not collected or included in our study. The guano pellets of these smaller bat species were smaller than pallid bat guano and were never stained red like the pallid bat scats. For pallid bats, mean scat diameter = 3.1 mm, mean length = 7.8 mm n = 23); for Townsend’s big-eared bats, mean diameter = 1.9 mm, mean length = 3.8 mm (n = 7); for cave myotis, mean diameter = 2.0 mm, mean length = 4.0 mm (n = 14). The Mexican long-tongued bat, C. mexicana (10–25 g), also used soil-piping cavities at CCNP, but it was never found roosting in the same cavity as pallid bats. The Mexican long-tongued bat is a specialized nectar and pollen feeding bat whose guano lacks visible insect fragments, is primarily composed of pollen sometimes with bits of anthers and filaments from the stamens, and forms yellowish or reddish-brown splats beneath its roosts rather than pellets, similar to that of other nectar-and-pollen feeding bats (W Peachey and N Czaplewski, pers. obs., 1987). Large, red-stained guano pellets exactly like the pallid bat scats in the soil-piping cavities accumulated on plastic sheets laid beneath the roost on the porch of the caretaker’s building, indicating that at times, both groups of pallid bats were feeding on mesquite bugs.

Only one non-insect prey item was found beneath the pallid bat night roosts, a partial bat wing with metacarpals II-III-IV, phalanges, and a bit of attached membrane of the wing tip. The proximal ends of the metacarpals are morphologically distinct from those of the vespertilionid bats of the Cienega Creek area, and represent those of the phyllostomid, Choeronycteris mexicana. The distal ends of the metacarpals and the phalanges have the epiphyses completely fused, indicating an adult bat. The skin attached to the wing bones showed some signs of feeding by decomposer arthropods, indicating that the wing had been beneath the pallid bat roost for some time before it was collected in February 2004.

Figure 3 Stereopair of bat teeth and insect fragments.

(A, B), Stereopair photograph of the upper teeth and anterior palate of a skull of Antrozous pallidus (anterior is toward the top of the image) showing the robust upper canines with strong longitudinal flanges, which help to penetrate and puncture thick chitin. Incisors and premolars are also visible. (C–I). Pieces of the exoskeletons of insects discarded by A. pallidus, showing tooth punctures caused by the bats. (C) Elytron of a beetle Chrysina gloriosa (Scarabaeidae); (D) Same as (C), close-up of area enclosed by red rectangle in (C), rotated 90° counterclockwise and enlarged to show tooth punctures. (E) elytron of a dung beetle Dichotomius colonicus (Scarabaeidae). (F) Hind leg of mesquite bug Thasus neocalifornicus (Coreidae). (G) Elytron of Cyclocephala (Scarabaeidae). (H) Elytron of Xyloryctes thestalus (Scarabaeidae). (I) Head, thorax, and partial elytra of darkling beetle Stenomorpha marginata (Tenebrionidae). Scale bar in each image is in mm. Photos by NJ Czaplewski.

Pallid bats foraging in and around the mesquite bosque clearly used the soil-piping cavities as a place to hang and process insects they catch. Pallid bats are equipped with robust jaws and teeth for their body size, including longitudinally curved, tapered canines with four heavy crests or flanges on the anterior, lingual, posterior, and labial surfaces running from the apex to the base of the tooth crown, with deep furrows between all except the anterior and labial flanges (Figs. 3A–3B). These canines are adapted for procuring and puncturing the exoskeletons of hard-bodied insects. The flanges of the canines create stress and propagate cracks in the chitin, making it easier to penetrate the exoskeleton (see Freeman, 1979; Freeman, 1992; Freeman, 1998; Freeman & Weins, 1997) and subdue an insect. The tooth marks of the bats are readily seen on many of the culled fragments (Figs. 3C–3I).

At Cienega Creek, pallid bats fed on at least 36 species of large insects (approximately 25–60 mm body length) based on parts discarded beneath the night roosts (Table 1). Of these insects, 17 taxa are reported for the first time in the diet of A. pallidus. No arthropod groups other than insects were represented. All exoskeletal parts appeared to be those of adult insects. We found no evidence that the pallid bats fed upon the noxious nymphs of mesquite bugs. The insects eaten include mainly night-active forms, many of which are ground dwelling, although a few diurnal taxa including several grasshoppers, two long-horned beetles, and a dragonfly were taken.

Table 1 List of insects and one bat identified from culled body parts deposited beneath pallid bat roosts at Cienega Creek, Arizona.

Hemiptera	Coreidae	Thasus neocalifornicus mesquite bug	1,303 (88%)	
Coleoptera	Scarabaeidae	Chrysina gloriosa glorious scarab	1	
		Polyphylla decemlineata ten-lined June beetle	4	
		aStrategus aloeus ox beetle	1	
		Strategus sp. ox beetle	12	
		aXyloryctes thestalus rhinoceros beetle	1	
		Cyclocephala sp. masked chafer	1	
		aDichotomius colonicus dung beetle	1	
		Cotinis mutabilis green fig beetle	2	
		Tomarus sp. carrot beetle	1	
		Phyllophaga sp. May beetle	1	
	Tenebrionidae	aStenomorpha marginata darkling beetle	2	
		Stenomorpha sp. darkling beetle	1	
		Eleodes sp. Pinacate or darkling beetle	2	
	Carabidae	Calosoma scrutator fiery searcher	4	
		Pasimachus sp. ground beetle	1	
	Hydrophilidae	Hydrophilus sp. giant black water beetle	1	
	Cerambycidae	Oncideres rhodosticta mesquite girdler	1	
		aPrionini long-horned beetle	1	
Orthoptera	Tettigoniidae	Microcentrum rhombifolium greater angle-wing katydid	2	
		aNeoconocephalus triops broad-tipped conehead	17 (1.1%)	
		aScudderia mexicana Mexican bush katydid	6	
	Acrididae	Schistocerca nitens gray bird grasshopper	76 (5%)	
		Melanoplus differentialis differential grasshopper	4	
		aPhlibostroma quadrimaculatum four-spotted grasshopper	1	
		aTrimerotropis cyaneipennis blue-winged grasshopper	1	
Lepidoptera	Sphingidae	Hyles lineata white-lined sphinx	6	
		aEumorpha vitis vine sphinx	11	
		aSphinx sp. sphinx moth	4	
		Manduca sexta tobacco hornworm moth	1	
	aTortricidae	Indeterminate leafroller moth	1	
	Noctuidae	Catocala sp. underwing moth	1	
	Saturniidae	aSphingicampa ( =Syssphinx) hubbardi mesquite moth	1	
		aAutomeris iris iris-eyed silkmoth	1	
Neuroptera	Myrmeleontidae	aVella fallax antlion	2	
Odonata	aAeshnidae	Indeterminate darner	1	
Blattodea	Corydiidae	Arenivaga sp. cockroach	1	
Diptera	Tipulidae	aNephrotoma sp. tiger crane fly	1	
Chiroptera	Phyllostomidae	aChoeronycteris mexicana Mexican long-nosed bat	1	
Notes.

a Indicates new record of prey consumed by pallid bats. Fourth column shows number of identified body parts and percentage of total when over 1%.

In terms of relative abundance, the vast majority of insects consumed by pallid bats at all three cavity roosts were adult mesquite bugs. This insect also accounted for many of the bat guano pellets being reddish. Of 483 total identified insect parts, 429 (88.8% frequency) were of mesquite bugs. All body parts of the mesquite bugs are represented, but mostly the least nutritious and most chitinous portions (wings, legs, antennae) were discarded; relatively few abdomens were found beneath the bat roosts (Table 2). Thus, pallid bats were eating mainly the abdomens of the mesquite bugs. Interestingly, the relatively few available remains of Thasus abdomens showed that the softer, ventral portion was selectively eaten and the remainder of the abdomen discarded. Of the identified Thasus parts, 272 forewings (Table 2) indicate a minimum of 136 individual mesquite bugs eaten. In most samples there were more male than female mesquite bug hindleg elements, although in one sample there were more female than male hindleg elements. For insect species other than mesquite bugs, relative abundance was low, representing only one to four individuals of most species. One exception to this was the gray bird grasshopper, Schistocerca nitens, represented in February 2002 by 19 forewings and 57 hindwings.

Table 2 Body parts of adult mesquite bugs (Thasus neocalifornicus) discarded by night-roosting pallid bats and collected in soil-piping cavities in Cienega Creek Natural Preserve on three visits between January 2001 and September 2002, in decreasing order of abundance.

Body parts	Number of elements collected	
	January 2001	February 2002	September 2002	
Forewings	272	213	127	
Leg parts, total	183	43	99	
Forelegs and midlegs	91	–	21	
Hind tibias	52 (12 F, 40 M)	22 (8 F, 14 M)	57 (21 F, 36 M)	
Hind femurs	40 (13 F, 27 M)	21 (7 F, 14 M)	31 (20 F, 11 M)	
Hindwings	40	45	56	
Isolated antennae	9	–	0	
Thorax (dorsal portion)	8	0	18	
Heads with attached antennae	5	1	8	
Abdomens	3	8	5	
Thorax with attached fore- and hindwings	1	1	0	
Notes.

F female

M male

Discussion

Ross (1967) and subsequent authors have compiled a long list of arthropod prey species taken by pallid bats. Our results add 17 taxa not previously recorded as pallid bat prey to the overall list. Pallid bats in our study fed upon large moths as well as large beetles; Freeman & Lemen (2007) indicated that beetles were about 3.2 times harder than moths of the same body size, but that body size or volume of the insect also was important in cuticle toughness. Freeman & Lemen (2007) hypothesized that, as aerial feeders, some bats must limit the upper size of insects they eat, because insects that are too large cannot be processed orally in flight, especially for a bat species that depends on being able to continue echolocating to fly. Some bats might capture prey that are too tough to process in flight and must land to process them. These authors also hypothesized that harder insects might take longer for bats to chew and thus limit the upper size of certain taxa of insects taken, which varies among insect taxa.

In our study, the higher numbers of large, armored, and cumbersome legs and other body parts of mesquite bugs found beneath roosts relative to other taxa of insects suggests that mesquite bugs are more difficult for pallid bats to process than other kinds of insects. Mesquite bugs have a small head, thorax, and abdomen with large legs relative to most of the beetles and moths in the bats’ diet. Perhaps the relative ease with which mesquite bugs are located or secured in late summer or early autumn counterbalances the energy and time needed to commute to a night roost to process them. The size and hardness of the insects eaten by pallid bats suggests there is a large upper size limit to what insects pallid bats are capable of processing and eating. Mesquite bugs (Thasus) are among the largest terrestrial heteropterans known (Forbes & Schaefer, 2003) and are 28–43 mm in body length. The relative abundance of their body parts recovered in the soil-piping cavities leads us to hypothesize that the availability of the cavities and their proximity to the mesquite bosque enhanced their usage by pallid bats as a place to process a seasonally abundant source of food.

As noted above, most of the insects eaten by the pallid bats are nocturnal, although several species represented in our study are diurnal. Most day-active insects are inactive at night. Therefore, for bats that must be able to hear prey-generated sounds of motion to find prey, our prey list largely supports the assertion of Fuzessery et al. (1993) that pallid bats are hunting primarily with sound cues and are less dependent on visual cues. Many of the large insects consumed by pallid bats in this study make noise in flight, while others have been variously described as noisy fliers (e.g., Cotinis mutabilis; Tallamy, 2009).

Furthermore, in spiny mesquite foliage, the bats probably avoid flying in the understory or within the tree canopy to hunt for prey they cannot hear. The risk of injury is too high unless they can be certain there is potential food available there, like katydids, mesquite moths, and mesquite bugs.

As adults, mesquite bugs do not secrete the same compounds as a defense against insect predators that they do as nymphs. Most insect predators are not interested in the non-toxic adults because the bugs are so big. For the mesquite bugs, it might not be evolutionarily worthwhile to invest energy in producing toxic compounds against other insect predators when it is unnecessary. The adult bugs switch to a defense of muted colors (thus being more cryptic to visually-oriented aerial predators), and a physically more armored exoskeleton (spiny hind legs, tougher wings), but less noxious chemicals than nymphs. We hypothesize that this is not necessarily a change to prevent predation as much as a trade-off of putting less investment in defense (producing energetically expensive coloration and toxins) and more investment into reproduction (wings provide mobility to find mates, less toxic chemical investment for short period of mating and death).

As noted earlier, in laboratory experiments pallid bats showed an aversion to the odor of a Pinacate beetle Eleodes (Johnston, 2002); however, at least one species of Eleodes, E. acuticauda, has been reported as a prey item for pallid bats (Orr, 1954:232), and the genus also appeared as prey in our study. Perhaps the bats are able to process and discard the noxious parts of certain insects as well as the armored hard parts. Although the chemicals secreted by adult mesquite bugs differ from those of nymphs (chemical components frequently change after metamorphosis; Noge, 2015), the adult compounds have not been tested with vertebrates, so it is unknown whether the chemicals produced by the adult bugs actually deter vertebrate predators (Prudic, Noge & Becerra, 2008). Of the secretions produced by the adult bugs (hexyl acetate, hexanal, 1-hexanol and possibly others), hexyl acetate and hexanal might be aggregational pheromones directed toward other mesquite bugs (Prudic, Noge & Becerra, 2008; Noge, 2015). Given that pallid bats in our study never ate nymphs, the nymphal secretions might be effective not only against insect predators but also against bats. And given the frequency with which pallid bats ate the adults, either the adult bugs are non-noxious to pallid bats or else the bats are not susceptible or averse to their secretions.

The only non-insect prey item found at a pallid bat night roost in this study was another bat, the flower-visiting phyllostomid Choeronycteris mexicana. As noted above, C. mexicana utilized separate but adjacent soil-piping cavities at CCNP in summers during our study. There is one previous record of pallid bats eating a Mexican free-tailed bat, Tadarida brasiliensis, although the predation occurred while the two species were in captivity, being held together in the same cage from which the smaller free-tailed bats were unable to escape (Engler, 1943). Thus, the C. mexicana at CCNP is the first recorded instance in the wild of predation by A. pallidus on another species of bat.

Many kinds of animals take advantage of situations arising as they acquire food (Young, 2012). Like many predators, various species of bats are opportunistic on hatches of insects (e.g., Myotis [Vespertilionidae], Fenton & Morris, 1976; Lavia frons [Megadermatidae], Vaughan & Vaughan, 1986; Dial & Vaughan, 1987; Taphozous melanopogon [Emballonuridae], Hipposideros sp. [Hipposideridae], and Scotophilus temminckii [Vespertilionidae], Gould, 1978; Hipposideros gigas [Hipposideridae], Vaughan, 1977; Nycteris grandis [Nycteridae], Fenton et al., 1993) and also passively use sounds produced by the insects rather than actively echolocating them. Several of the large insects preyed upon are noisy fliers, and pallid bats might thus detect them easily. Some, like antlions, are poor fliers as adults (Merlin, 2003). At CCNP, opportunistic feeding was associated with high selectivity for a single prey species that could make wide searches for patches of food energetically worthwhile.

The body parts discarded versus parts eaten indicates that pallid bats take the most easily digestible and probably most nutritious parts of mesquite bugs. The abdomens of gravid female bugs filled with egg masses in particular might provide additional protein. When mesquite bugs are clustered and immobile on the periphery of the mesquite canopy, bats can likely capture the bugs easily compared to within the thorny canopy. Mated females move from the periphery deeper into the mesquite tree canopy to find appropriate places to deposit their eggs, and thus are less susceptible to being located and preyed upon compared to males, which might explain the male bias in our samples. The noxious and aposematic defenses of the nymphs, which are unable to fly, render them relatively immune to attack at night by the bats perhaps due to their odor and quieter movements, and to visually-oriented predators like birds during the day (or bats during twilight). By becoming immobile overnight after they alight on mesquite foliage at evening twilight, the bugs might avoid detection by bats. The noisy flight and possibly other movements and activities of the mesquite bugs, beetles, grasshoppers, and other large insects has been little studied and could be an important aspect of the bat-insect relationship. Similarly, the influence of anthropogenic noise (e.g., automobile traffic, railroad noise, air traffic) on a passive-sound-using predator limits the pallid bats’ foraging efficiency and potentially their ability to utilize certain areas for foraging (Bunkley & Barber, 2015; Bunkley et al., 2015).

Unfortunately, in this study we were unable to collect data seasonally or regularly, but a seasonal or monthly collection of dietary data would provide a good future study to pursue this ecological relationship in greater depth. Moreover, the bats might select mesquite bugs as prey only when the bugs are the most vulnerable: in late summer or early autumn after the adult females laid the eggs for the overwintering generation, and when falling nighttime air temperatures, local cool air drainage from the adjacent mountains and foothills, and high evapotranspiration might slow the insect’s activity or mobility. Finally, the overwhelming majority of insects consumed in the night roosts were mesquite bugs locally derived from the mesquite bosque; the bats consumed other kinds of insects almost incidentally. In addition to their ability to endure injuries and heal (Davis, 1968), the dietary plasticity shown by pallid bats across the species’ broad geographic range might help to lessen their risk of extinction (Boyles & Storm, 2007) in the face of anthropogenic environmental upset and climate change.

Conclusions

The diet of pallid bats can be investigated non-intrusively by visiting their temporary-use night roosts during the day while the bats are away at separate day roosts. However, the night roosts possibly yield evidence only of those foods that are large enough to require transport to a temporary night roost for processing of edible versus inedible parts. Adult mesquite bugs formed the predominant prey for pallid bats at the CCNP. We found no evidence of pallid bats feeding on toxic, aposematically colored nymphal stages of mesquite bugs. Adult mesquite bugs are possibly non-toxic to pallid bats, or perhaps the bats are able to tolerate the less-toxic compounds of the adult bugs. Late-season breeding and postbreeding adult mesquite bugs are exposed near the edges of the mesquite canopy and provide prey for opportunistic, foliage-gleaning pallid bats. After breeding and laying eggs that overwinter in the mesquite trees, moribund adult mesquite bugs begin to become immobile in the trees or drop from the canopy when the nighttime low temperatures at CCNP fell below 21 °C. Mesquite bugs are considered to be mostly subtropical-tropical insects that may have invaded the southwestern United States during historic times with the bringing of cattle and spread of mesquite trees; pallid bats at the CCNP are providing an important natural control on the local mesquite bug population. Their exploitation of mesquite bugs is probably enhanced by the proximity of soil-piping caves to the mesquite bosque as a place to process the insects. Pallid bats at the CCNP ate numerous taxa of large-bodied insects, consistent with their diet in many other portions of the bats’ range. When mesquite bugs are observable in the local mesquite trees, their procurement by pallid bats can be determined by the presence of large reddish guano pellets 2.5–3.5 mm in diameter beneath local bat night roosts. Insects parts discarded beneath pallid bat roosts can be distinguished from insect parts culled by birds or other predators by distinctive tooth marks on the discarded insect parts. Bats usually ate the abdomen and thorax of mesquite bugs and most consistently discarded the wings and legs. At the CCNP, pallid bats left the remains of no arthropods other than insects. Seventeen taxa of insects were newly identified as prey for pallid bats, and reflect a diversity of local habitats of the CCNP as foraging habitat for the bats. In addition, pallid bats ate an individual of one other local species of bat, the Mexican long-nosed bat, another first recorded instance of such predation for pallid bats.

Supplemental Information

Figure S1 Raw data (camera lucida) drawing showing how to distinguish female from male mesquite bug hind femurs and hind tibias

Click here for additional data file.

Table S1 Raw data measurements of bat guano pellets

Click here for additional data file.

Supplemental Information 1 Thasus (mesquite bug) defensive chemicals and online sources

Click here for additional data file.

Supplemental Information 2 Raw data listing field observations and identifications of insects processed and eaten by Antrozous pallidus in soil-piping cavities at Cienega Creek Natural Preserve, Arizona

Click here for additional data file.

We thank the insects and the pallid bats themselves for providing inspiration and endless actions to pique our curiosity. We thank Richard Packauskas for aid in the initial identification of the mesquite bug, Gene Hall, University of Arizona, for identifying a small initial collection of insect parts, Tom Bethard, Kevin Horstman, Samantha Lefevre, and Robert Pape for aid in fieldwork, Jessica Czaplewski for help in sorting insect parts, Cheryl D Czaplewski for logistical support, and Julia Fonseca, Pima County Flood Control, for hydrological information about soil-piping. Thanks to Gale Bundrick and the Pima County Parks and Recreation Department for permission to access the area for study. We thank Melissa Sadir of the Collection of Recent Invertebrates at the Oklahoma Museum of Natural History for accessioning and cataloging insect parts. We appreciate the loan of a pallid bat specimen by Brandi S. Coyner and Janet K. Braun of the Collection of Mammals at the Oklahoma Museum of Natural History. We thank Steve Westrop and Roger Burkhalter for the use of their bellows camera with Stackshot rig and focus-stacking software to capture and process images. We appreciate the many contributors to the websites: Arizona: Beetles Bugs Birds and more (Margarethe Brummermann), BugGuide.net, ButterfliesandMoths.org, The Moths of Southeastern Arizona, and Wikipedia for their time and efforts in providing the ease of access and helpful information that proved highly useful in researching habitat and food plants for many of the insects.

Additional Information and Declarations

Competing Interests

Author Contributions

Animal Ethics

Field Study Permissions

Data Availability

The authors declare there are no competing interests. Nicholas J. Czaplewski and Katrina L. Menard are curators at the Oklahoma Museum of Natural History, a part of the University of Oklahoma. William D. Peachey is an Independent Researcher and member of Sonoran Science Solutions, a not-for-profit, all volunteer, research, conservation, and education organization.

Nicholas J. Czaplewski and William D. Peachey conceived and designed the experiments, performed the experiments, analyzed the data, contributed reagents/materials/analysis tools, prepared figures and/or tables, authored or reviewed drafts of the paper, approved the final draft.

Katrina L. Menard performed the experiments, analyzed the data, contributed reagents/materials/analysis tools, prepared figures and/or tables, authored or reviewed drafts of the paper, approved the final draft.

The following information was supplied relating to ethical approvals (i.e., approving body and any reference numbers):

By design, in order to avoid disturbance to the bats, we chose a noninvasive method and collected insect remains at pallid bat night roosts that were not used by the bats during the daytime when we visited. (The bats day-roosted elsewhere.) No contact was made with live vertebrates during the study. Thus, IACUC approval was not required because we collected only discarded prey parts.

The following information was supplied relating to field study approvals (i.e., approving body and any reference numbers):

The Arizona Game and Fish Commission provided permit no. I-96-024. We received oral permission to conduct fieldwork on the Cienega Creek Natural Preserve property from Gale Bundrick of the Pima County Parks and Recreation Department, Tucson, Arizona.

The following information was supplied regarding data availability:

The data is available in Tables 1 and 2. Insect fragments are in the process of being accessioned and cataloged into the Recent Invertebrates Collection of the Oklahoma Museum of Natural History. Once catalogued, the relevant data will be provided through the GBIF and iDigBio online portals.

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
