# Peer review of "Mesquite bugs, other insects, and a bat in the diet of pallid bats in southeastern Arizona"

_PeerJ, doi:10.7717/peerj.6065_

## Round 0.1 · original submission · Minor Revisions

· Academic Editor

Minor Revisions

I have received two expert reviews on your MS. As you can see, the reviewers had quite disparate opinions of your work with Reviewer 1 questioning its significance. I tend to concur with Reviewer 2 that even a descriptive natural history study is worth publishing as in this case where it can extend what was previously observed. I appreciate the value in identifying novel prey species and in demonstrating flexibility in foraging. However, I do also agree with both reviewers that the paper is too long for what it contributes. Please focus your discussion on the specific novel findings here and their agreement with previous research. I agree with Reviewer 2 that you might emphasize the finding that pallid bats may have fed on another bat species. Try to be more explicit about any hypotheses and/or the novel contribution of the current data in the revision. Please try to address both reviewers' comments and concerns.

Reviewer 1 ·

Basic reporting

It is not clear the importance of this study.

The authors should show the number of collected insect parts in Table 1.

It may be better to describe the habitats of the identified prey in Table 1 (mesquite bosque or outside it).

“Toxic pheromone” or “noxious pheromone” is funny. Pheromones are compounds that are involved in intraspecific interactions. Pheromones are signals to conspecifics, and basically not toxic or noxious to conspecifics. On the other hand, defensive compounds are toxic, deterrent or repellent to predators. “Toxic pheromone” is funny because the term for intraspecific interactions is mixed with one for interspecific interactions.

Experimental design

I think the data in this study are preliminary.

The authors said that the mesquite bugs were immobile below 21 °C. How did they measure the mobility of this bugs?

Validity of the findings

Most of the discussion are not based on the results in this study. Discussion is too long.

Additional comments

This manuscript written by Czaplewski describe the diet of pallid bats by collecting inedible parts of prey. The authors found that the bats predominantly eat adult mesquite bugs in the experimental period. Because the mesquite bugs appear in a limited season, it is difficult to understand what the bats eat regularly. It is not clear what is the preferable prey of the bats in an area where the mesquite bugs are not distributed. Because the lack of these points, it is not easy to understand the importance of this study.

Reviewer 2 ·

Basic reporting

No comment.

Experimental design

No comment.

Validity of the findings

No comment.

Additional comments

"Mesquite bugs and other insects in the diet of pallid bats in southeastern Arizona"

This is a natural history note describing three samples of the diet of pallid bats from insect parts at night roosts.

The most interesting part to me was the first evidence that a wild pallid bat ate another bat. This finding should probably be in the title.

What is the evidence that a pallid bat ate the Choeronycteris and that this bat was not eaten by something else?

I have no major problems with this paper. It’s nice to see people writing natural history papers like this one.

The writing is very thorough, but it could be greatly improved by deleting unnecessary words. It could easily be cut in half. But this is a minor issue. To give one example, consider this randomly chosen sentence:

“Hind leg parts (hind femurs and hind tibias) allowed determination of the sex of the individual mesquite bugs eaten. Males have inflated hind femurs with projecting spines while those of females are not inflated and lack spines, and males have hind tibias ridged with a central bend and spur while females have a straight hind tibia without a spur (Schuh & Slater, 1995)”

This sentence could be:
“We inferred sex using sexually dimorphic hindlimbs.”

Many of the adverbs and adjectives could be deleted. Another way to omit words is to reduce some of the speculation. For example, line 540, “The uncommon occurrence of C. gloriosa remains (one elytron) beneath the bat roosts indicates that the pallid bats move away from the mesquite bosque at times to hunt along the riparian vegetation of the creek and on adjacent uplands among the occasional junipers.” Can we really infer all that from a single beetle wing?

Much of the information is also stated several times.

The authors should avoid providing links to websites. These links will change with time, e.g. Lines 111, 531, 447, 454.

Other minor comments:
Lines 17 and 19 need revision. There are words missing.

Line 32 should maybe say “possible predation” or say why it’s certain to be the pallid bat who ate the other bat.

The level of precision on numbers lines 316, 368 is too high for such a small sample.

Line 558. What is the sentence saying? In a refuging species like the pallid bat, efficient and rapid dispersal and the ability to exploit patchy food resources is probably essential. Maybe delete.

---

## Round 0.2 · Minor Revisions

· Academic Editor

Minor Revisions

The paper is still lacking focus. The opening pages read like a descriptive essay but without clear hypotheses or problem statements. You need to be more focused in terms of “previous research has shown.. Our observations indicate…” so that it is clear how you are furthering or modifying existing knowledge. There is a bit of a lack of organizational structure to the introduction. It should be clear from the outset what the aims of the MS are. The reader should not have to wait until the final sentence of the introduction.

The writing still needs some work. Comments refer to the tracked changes MS. For example, please insert commas after clauses. E.g., after “In a few areas” on line 24 and after “Until recently” on line 25. Please check throughout.

The abstract could be shortened. You don’t need as much detail about the bats’ habitat and foraging ecology here, or the ecology of mesquite bugs. State simply that these bats were thought to primarily consume insects, but your observations indicate greater dietary flexibility (or whatever you think the key message should be). Be focused on the key findings. Avoid speculation that you did not test (e.g., lines 46-48). In addition, you do not observe the bats across various habitats or seasons so you really cannot speak to plasticity as a function of diverse habitats or seasonal changes. Save speculation for the discussion, but, in the abstract, stick to key methods and findings.

Your writing could still be more concise. As just one example, “Studies of their dietary habits..” could just be written as “Studies have shown primarily predation on insects…” (lines 61-63). Delete “used this method” on line 232.

I do not understand line 94 as written.

It is not clear how the observations of mesquite bug behavior factor into your analysis (lines 254-260). If you are attempting to demonstrate that changes in insect behavior affect the bats’ foraging strategies, make this connection explicit. Stick to hypotheses that can actually be addressed with your data. I think the data will make the most impact if you can show how the overall composition of food or bats’ habits change along with changing density of mesquite bugs (for ex.). As the study is written in a purely descriptive manner, it is difficult to extract important patterns.
The discussion is still far too long. Please try to cut by at least 1/3. Please stick to information that is pertinent to your own observations.

---

## Round 0.3 · accepted · Accept

· Academic Editor

Accept

Thank you for your attention to the final details from the last round of review.

#